# Integrative Analysis of Metabolome and Transcriptome Identifies the Role of γ-Glutamylcysteine in Mitigating Deoxynivalenol-Induced Toxicity

**DOI:** 10.3390/toxins17090457

**Published:** 2025-09-12

**Authors:** Xiaocheng Bao, Xiaolei Chen, Shuai Chen, Ming-An Sun, Hairui Fan

**Affiliations:** 1Institute of Comparative Medicine, College of Veterinary Medicine, Yangzhou University, Yangzhou 225009, China; mx120240996@stu.yzu.edu.cn (X.B.); chenshuai@yzu.edu.cn (S.C.); mingansun@yzu.edu.cn (M.-A.S.); 2College of Animal Science and Technology, Yangzhou University, Yangzhou 225009, China; mx120230917@stu.yzu.edu.cn

**Keywords:** deoxynivalenol, metabolome, transcriptome, toxicity, γ-glutamylcysteine

## Abstract

Deoxynivalenol (DON), a prevalent environmental toxin produced by Fusarium fungi, frequently contaminates feed and food products. However, the critical metabolites and regulatory factors involved in DON toxicity remain poorly understood. Building upon our established DON-induced porcine intestinal epithelial cells (IPEC-J2) injury model, this study employed liquid chromatography-tandem mass spectrometry (LC-MS/MS) to conduct metabolomic analysis, and integrated analysis with transcriptomic data from DON-exposed IPEC-J2. Results identified 1524 differentially expressed metabolites, and revealed significant enrichment in Glutathione metabolism and Mucin-type O-glycan biosyn-thesis pathways. Notably, γ-glutamylcysteine (*γ*GC), the rate-limiting precursor for glutathione synthesis, showed significant reduction following DON exposure. To explore the biological function of *γ*GC, this study found through exogenous supplementation experiments that *γ*GC pretreatment could significantly alleviate the inhibition of IPEC-J2 cell viability, effectively reduce intracellular ROS accumulation and inhibit DON-induced apoptosis in IPEC-J2 cells. These results indicated that the severe oxidative stress induced by DON is closely related to the blockage of glutathione synthesis caused by the exhaustion of intracellular *γ*GC, and revealed the application potential of *γ*GC as an exogenous supplement in the prevention and treatment of DON exposure. In conclusion, this study offers valuable insights into the metabolic and transcriptional alterations, along with the key metabolites and regulators involved in the cellular response to DON pollution. It also lays a theoretical foundation for more effective prevention and treatment strategies against DON pollution.

## 1. Introduction

Deoxynivalenol (DON, Vomitoxin) is a secondary metabolite produced by fungi such as Fusarium species, which is widely present in crops and related products, especially in grains and feeds, presenting a significant risk to both human and animal health [1]. Under unfavorable environmental conditions, such as high temperature and humidity, fungi may grow, reproduce, and release toxins during the cultivation or storage of grains [2]. Among various domestic animals, pigs are regarded as the species most sensitive to DON. Even low levels of DON exposure may impact animal health, leading to various pathophysiological symptoms, such as anorexia, vomiting, diarrhea and gastrointestinal necrosis [3,4]. Over the 10-year period from 2008 to 2017, a total of 74,821 feed and feed raw material samples were collected globally, and the contamination rate of DON was found to be as high as 64% [5]; however, a recent report on the detection of mycotoxins in feed samples collected in China between 2018 and 2020 revealed that the contamination rate of vomitoxin exceeded 96.4%, with average concentrations ranging from 458.0 to 1925.4 µg/kg [6]. Studies have shown that DON can be efficiently absorbed in the small intestine of pigs, with only a small portion being converted into the low-toxicity metabolite DOM-1 in the colon. The vast majority of DON is excreted in its original form, glucuronic acid-DON and glucuronic acid-DOM-1 through urine [7]. This metabolic characteristic further amplifies the toxic effects of DON, leading to a lower tolerance of it in pigs. The intestinal tract of pigs is the primary target organ of DON’s attack. Previous studies have shown that DON exposure can disrupt the biological, physical and immune barriers of the intestinal tract, leading to a significant decrease in the expression levels of mucins and tight junction proteins (such as claudin and occludin), thereby damaging the integrity of the intestinal barrier and making it easier to induce the invasion of other pathogenic microorganisms [8,9]. At the molecular level, a previous study indicated that DON and its derivatives may damage cells by binding to ribosomes [10], thereby inhibiting protein synthesis, interfering with cell proliferation, suppressing the activation of various protein kinases, and disrupting normal gene expression [10,11,12]. Studies have shown that a diet containing 4 mg/kg of DON can significantly activate the MAPK signaling pathways in the jejunal and ileal tissues of weaned piglets [13]. The use of MAPK pathway inhibitors can significantly alleviate the cytotoxicity of DON on pig small intestinal epithelial cells (IPEC-J2) [14], inhibitors targeting NF-κB also alleviate the cell damage caused by DON [15]. However, DON has strong chemical stability and is not easily degraded during feed processing, transportation and storage, thus maintaining its biological toxicity continuously [16]. At present, various anti-mold and de-mold methods have been developed, including physical adsorption, chemical modification and biological degradation, etc. However, all these strategies have problems such as low degradation efficiency, poor stability or high application cost, and make it difficult to fundamentally solve the DON pollution problem [17]. Therefore, a thorough investigation into the toxic mechanism of DON-induced damage to pig intestinal cells and the defense mechanism of host cells will provide a mechanistic rationale.

Metabolomics, as an important technique for studying the overall metabolic state of biological systems, can comprehensively identify and quantify the dynamic changes in endogenous metabolites and has been widely applied in the research of disease mechanisms and toxicology [9,18,19]. Meanwhile, transcriptomics can be used to reveal the differences in gene expression under pathological conditions and assist in analyzing the potential upstream regulatory mechanisms induced by exogenous toxins [11,12]. In recent years, the combination of metabolomics and RNA sequencing (RNA-seq) has been proven to effectively identify metabolite changes and related differentially expressed genes (DEGs) caused by mycotoxin exposure or viral infection, providing a powerful tool for revealing toxicological mechanisms [20,21]. Disturbances in the metabolic state of cells not only directly affect functions such as cell division and apoptosis, but can also be utilized by viruses or toxins to regulate the physiological processes of the host. However, the specific mechanism of metabolic regulation of intestinal epithelial cells under DON exposure remains unclear.

Glutathione (GSH) is a key antioxidant within cells, maintaining REDOX homeostasis by scavenging free radicals and peroxides [22]. Its depletion is regarded as a key link in oxidative stress-induced intestinal inflammation and other diseases [23,24]. Although direct supplementation of GSH is regarded as a candidate approach for restoring antioxidant capacity, its clinical application still faces challenges due to the low transmembrane transport efficiency of GSH and the limitations of de novo synthesis, especially the rate-limiting step generated from γ -glutamylcysteine (γ-Glu-Cys, *γ*GC) [25]. *γ*GC, as a direct precursor of GSH synthesis, has good cellular uptake, can effectively bypass transmembrane transport barriers, and promote GSH synthesis. It is considered a potential target for regulating oxidative stress and improving GSH depleting-related diseases [26]. However, the changes in *γ*GC in the oxidative stress of pig intestinal cells induced by DON and its potential regulatory mechanisms have not been reported yet.

In this study, in an attempt to address the aforementioned issues, we employed non-targeted liquid chromatography-tandem mass spectrometry (LC-MS) together with transcriptome sequencing technology to systematically analyze the metabolic and transcriptional changes in IPEC-J2 intestinal epithelial cells after DON exposure. The results showed that DON significantly reshaped the metabolic characteristics of cells, induced cytotoxicity, and led to significant changes in multiple metabolites and gene expressions. In particular, we identified a group of differential metabolites-*γ*GC closely related to the toxicological process of DON, and verified their protective effect in alleviating DON-induced cell damage. This discovery not only expands the understanding of the toxicological mechanism of DON, but also lays a solid theoretical foundation for the future development of nutritional intervention strategies to prevent and control the toxicity of DON.

## 2. Results

### 2.1. Metabolomic Analysis in IPEC-J2 Cells Exposed to DON

By constructing the partial least squares discriminant analysis (PLS-DA) model, the relationship between metabolite abundance and sample category was clarified. The results showed that in both ionic modes, the samples in the DON treatment group and the control group could be clearly divided into two independent clusters in the metabolic characteristic space (Figure 1A), suggesting that DON exposure significantly reshaped the cellular metabolic profile. Further analysis of differential metabolites revealed that in the positive-ion mode, a total of 1202 differential metabolites were identified, among which 765 were up-regulated and 437 were down-regulated. Under the negative-ion mode, a total of 322 differential metabolites (DEMs) were identified, among which 173 were upregulated and 149 were downregulated (Figure 1B, Appendix A). To reveal the biological significance of these differential metabolites, we conducted a metabolic pathway enrichment analysis on them. It was also found that the differential metabolites were enriched in significantly different signaling pathways under the two positive-ion modes. Among them, the positive-ion mode was significantly enriched in Thiamine metabolism, Taste transduction, the Sulfur relay system, Sulfur metabolism, Sphingolipid metabolism, etc., were significantly enriched in positive-ion mode. Tyrosine metabolism, Taurine and hypotaurine metabolism, Taste transduction, Pyruvate metabolism and signaling pathways such as Purine metabolism were significantly enriched in negative-ion mode (Figure 1C, Appendix A). These results indicate that DON exposure induces significant alterations in multiple key metabolic pathways in IPEC-J2 cells, suggesting that they may play a core regulatory role in the toxic response.

### 2.2. Integrated Analysis of Metabolome and Transcriptome of DON-Exposed Cells

To systematically reveal the molecular mechanisms triggered by DON exposure and the involved signaling pathways, this study conducted an integrated analysis of DEMs and differentially expressed genes (DEGs) between the DON treatment group and control group, and simultaneously carried out pathway enrichment analysis. It is notable that in the glutathione metabolic pathway, γ-L-glutamyl-L-cysteine (γ-Glu-Cys, *γ*GC), is significantly downregulated after DON exposure (*p* < 0.0237, VIP = 1.4317) (Figure 2A, Appendix A). Furthermore, research indicates that *γ*GC is the direct precursor for the synthesis of Glutathione (GSH), and GSH, as the most significant antioxidant within cells, plays a crucial role in maintaining the REDOX balance of cells, eliminating reactive oxygen species (ROS), and protecting DNA, proteins, and lipids from oxidative damage. *γ*GC is regarded as the decisive factor restricting intracellular GSH synthesis [27]. In this study, we observed that *γ*GC was significantly downregulated after DON exposure (Figure 2B). Combined with our previous research results, DON can induce a strong oxidative stress response in IPEC-J2 cells and inhibit the expression levels of related antioxidant enzymes (such as MDA, CAT, T-SOD), and this phenomenon is highly consistent [28]. These results suggested that DON exposure may block the glutathione metabolic pathway by inhibiting the synthesis of *γ*GC, thereby weakening the antioxidant capacity of cells and ultimately inducing oxidative damage and cytotoxic effects.

### 2.3. γGC Attenuated DON-Induced Cytotoxicity in IPEC-J2 Cells

To further clarify the potential protective effect of *γ*GC in DON exposure, we first evaluated the effects of different concentrations of *γ*GC on the viability of IPEC-J2 cells. The results showed that within the range of 50–500 μM, *γ*GC had no significant toxic effect on cell viability (Figure 3A). Then, we intervened by supplementing *γ*GC after the cells were exposed to DON. The experimental results indicated that *γ*GC significantly alleviated the cytotoxicity induced by DON within the concentration range of 100–500 μM and presented a dose-dependent trend (Figure 3B). The cell confluency of the DON + *γ*GC group was significantly higher than that in the only DON treat group (Figure 3C). Based on the above results, we ultimately chose to supplement 200 μM *γ*GC after DON exposure as the treatment condition for subsequent mechanism research.

### 2.4. γGC Attenuated DON-Induced Oxidative Stress in IPEC-J2 Cells

Previous studies have indicated that DON exposure led to severe IPEC-J2 cells oxidative stress [28]. Therefore, to further verify the protective effect of *γ*GC in DON-induced oxidative stress, we detected the changes in intracellular reactive oxygen species (ROS) levels. Our results showed that DON exposure significantly induced the accumulation of ROS about 18.27%, while the supplementation of *γ*GC significantly inhibited the increase in this ROS level by about 8.74% (Figure 4A). To further indicate that the *γ*GC attenuated DON-induced oxidative stress, the mitochondrial membrane potential (ΔΨm) in IPEC-J2 cells was measured after different treatments. Meanwhile, we quantified the alterations in ΔΨm, showed that DON exposure significantly decreased the ΔΨm compared with that of the control group, while co-treated with *γ*GC mitigated the effect significantly (Figure 4B). These results indicate that *γ*GC can effectively alleviate oxidative stress and protect cells from damage caused by DON toxicity by eliminating the accumulation of ROS induced by DON.

### 2.5. γGC Attenuated DON-Induced Apoptosis in IPEC-J2 Cells

To further understand the effect of *γ*GC on DON toxicity, we examined the effect of DON exposure after *γ*GC pretreatment of cells on the apoptosis level of IPEC-J2 cells. Flow cytometry was applied for detection, and the results showed that the level of DON-induced apoptosis was significantly reduced after *γ*GC pretreatment during DON exposure (Control vs. DON vs. DON + *γ*GC = 1.0 + 0.09 vs. 2.78 + 0.05 vs. 1.38 + 0.14) (Figure 5A). And then, we further analyzed the level of BCl2 and BAX. Our results showed that DON significantly induced apoptosis of IPEC-J2 cells as shown by the decrease in BCL2, increase in BAX and the ratio of BAX/BCL2, and the addition of *γ*GC significantly improved or even reversed the harm caused by DON exposure (Figure 5B–D). Our results show that *γ*GC treatment reduces DON-induced apoptosis and confirms its protective effect against DON-related cell damage.

## 3. Discussion

In this study, we carried out a metabolomics analysis of intestinal cells exposed to DON and identified several metabolites that showed significant changes. Emerging evidence indicates that active metabolites can affect various aspects of the omics landscape, and integrating these findings with other omics data can offer valuable insights into the underlying biological mechanisms [29,30]. Thus, our thorough analysis of metabolic and transcriptional data has uncovered the key pathways impacted by DON exposure. Additionally, it offers a systematic overview of the biological modules involved in regulating cellular toxic responses to DON.

Our research first identified the alterations in metabolites of intestinal epithelial cells upon DON exposure. The differential metabolites were enriched in Thiamine metabolism, Taste transduction, Sulfur relay system, Sulfur metabolism, Sphingolipid metabolism and Tyrosine metabolism, Taurine and hypotaurine metabolism, Taste transduction, Pyruvate metabolism and signaling pathways such as Purine metabolism. In other DON metabolomics studies, it has been observed that the differential metabolites and enriched metabolic pathways resulting from DON toxicity exhibit significant variation. For example, the toxic effects of DON on the intestinal tract of zebrafish are primarily mediated through alterations in 2-oxo-carboxylic acid metabolism, amino acid biosynthesis, carbon metabolism, and certain aspects of lipid metabolism [9]. The reason for this might be the differences in the differential metabolites. In our study, we found that the differential metabolite *γ*GC is a direct precursor of GSH and is significantly downregulated after DON exposure. It seems reasonable to speculate that exogenous *γ*GC enters cells and is enzymatically transformed by GSS, thereby increasing the intracellular GSH concentration to counteract DON-induced oxidative stress. Studies have shown that GSH depletion can lead to the accumulation of ROS and exacerbate cell damage [31]. As an indispensable antioxidant, glutathione maintains the REDOX balance within cells by eliminating free radicals and peroxides [22]. Can directly providing GSH here alleviate the cell damage induced by DON? The study found that the uptake of extracellular GSH is restricted by γ -glutamyl transpeptidase, and its de novo synthesis is also limited by two key rate-limiting steps: Glutamic acid and cysteine are converted to *γ*GC through the following action of γ-glutamylcysteine ligase, and then from *γ*GC to glycine, promoted by glutathione synthase [25]. *γ*GC is a direct precursor of GSH, which can bypass the thermodynamic barriers related to transmembrane transport and is easily absorbed by various cells to promote GSH synthesis. Therefore, *γ*GC is regarded as a target for treating some diseases [32]. Early experimental studies have demonstrated that intraperitoneal administration of γ-glutamylcysteine (*γ*GC) can effectively restore glutathione levels depleted in various organs [33], and exerts hepatoprotective effects against liver injury induced by iron overload [34]. Furthermore, *γ*-GC has been shown to ameliorate oxidative damage in cultured neurons and astrocytes in vitro, while increasing cerebral glutathione content in vivo [35]. In addition, *γ*GC confers protection against toxicity induced by cecal ligation and puncture (CLP) or lipopolysaccharide (LPS) in murine models, with its anti-inflammatory mechanisms involving the elevation of intracellular glutathione (GSH) levels and the reduction in reactive oxygen species (ROS) accumulation [25]. These findings collectively suggest that *γ*GC may exert protective effects against deoxynivalenol (DON)-induced oxidative stress and inflammatory responses in intestinal cells. However, to date, no studies have specifically investigated the protective mechanisms of *γ*GC in pathological damage caused by related mycotoxins.

In this study, it was also found that γGC has no significant toxic effect on cell viability, and γGC significantly alleviates the reduction in cell activity induced by DON within a specific concentration range. And it shows a dose-dependent trend. This result preliminarily indicates that *γ*GC plays a role in resisting DON exposure-induced cell damage in intestinal cells. The detection of ROS levels and apoptosis levels further confirmed that *γ*GC supplementation could inhibit the accumulation of ROS and apoptosis induced by DON exposure by increasing the antioxidant capacity of cells, and to a limited extent alleviate the intestinal cell damage induced by DON exposure. This result is consistent with the function of the reported *γ*GC. Moreover, our previous research indicated that DON exposure can significantly cause inflammatory responses in intestinal epithelial cells [36]. Therefore, we speculate that *γ*GC might play a role in DON-induced intestinal epithelial cell damage through its anti-inflammatory properties. Previous studies have shown that melatonin, owing to its potent antioxidant capacity, can mitigate DON-induced oxidative stress and markedly inhibit autophagy in intestinal epithelial cells [28]. These findings suggest that various compounds may modulate DON-induced oxidative stress through diverse mechanistic pathways.

## 4. Conclusions

In conclusion, this study confirmed that the metabolic characteristics of intestinal epithelial cells were significantly reshaped under DON exposure conditions. These alterations are closely related to the cytotoxicity induced by DON. In particular, the level of *γ*GC was significantly downregulated after DON exposure, which was highly correlated with the increase in intracellular oxidative stress levels. Further functional verification experiments demonstrated that exogenous supplementation of *γ*GC could effectively alleviate the oxidative stress response induced by DON. The study not only deepens the understanding of the toxicological mechanism of DON, especially its regulation of oxidative stress and metabolic homeostasis, but also offers a mechanistic rationale and experimental basis for the application of *γ*GC as a potential nutritional intervention in the prevention and mitigation of DON pollution-related toxicity.

## 5. Materials and Methods

### 5.1. Cell Culture and DON Treatment

The IPEC-J2 cells line was used to build the model of DON exposure, and the cells was incubated with Dulbecco’s Modified Eagle Medium (DMEM) containing 10% fetal bovine serum (FBS, Gibco, Grand Island, NY, USA), in 5% CO_2_ at 37 °C. IPEC-J2 cells were seeded in six-well/twelve-well plates at a density of 5 × 10^5^ cells /mL. According to our previous, the model was treated with 1 µg/mL DON (C15H20O6; Sigma-Aldrich, St. Louis, MO, USA) for 48 h [11]. DON was dissolved in DMSO (initial concentration of 2 mg/mL) and diluted in culture medium to achieve a final concentration of 1 µg/mL. For *γ*GC (γ-Glutamylcysteine, MCE, HY-113402, Shanghai, China) supplementation, IPEC-J2 cells were treated with 200–500 μM *γ*GC during DON exposure.

### 5.2. Metabolites Extraction and LC-MS/MS Analysis

To explore the metabolite changes involved in the response of IPEC-J2 cells to DON exposure, this study was based on the previously established DON-induced injury model of IPEC-J2 cells [11]. Subsequently, LC-MS/MS technology was used to conduct non-targeted metabolomics analyses on the samples of the DON treatment group and the control group in both positive ion and negative-ion modes. When the degree of cell fusion is 60–70% based on our experience with the procedure and observations, add 1 μg/mL of DON to the cells and culture for 48 h. Eight biological replicates in DON treated and Control IPEC-J2 cells were set up for metabolomics detection. The sample processing method referred to the article we published previously [20] and the LC-MS/MS (High-resolution mass spectrometer Q Exactive, Thermo Fisher Scientific, Waltham, MA, USA) were performed by BIG (Shenzhen, China).

### 5.3. Metabolomics Analysis

The Compound Discoverer 3.1 (Thermo Fisher Scientific, Waltham, MA, USA) software was used to process the LC-MS/MS data, mainly including peak extraction, peak alignment and compound identification. Data preprocessing, statistical analysis, metabolite classification annotation and functional annotation were carried out using the self-developed metabolomics R software package metaX (version 1.4.2) [37] and the metabolomics information analysis process. The original data of multiple variables is dimensionally reduced through PCA (Principal Component Analysis) to analyze the grouping, trends (similarity and difference within and between sample groups), and outliers (whether there are abnormal samples) of the observed variables in this dataset. And according to the KEGG pathway database (https://www.genome.jp/kegg/pathway.html, accessed on 5 May 2025) to identify the enriched functionals terms and pathway of metabolites. Differentially expressed metabolites were identified with fold change > 1.2, and VIP > 1, q-value < 0.05 between the two groups.

### 5.4. Integrated Analysis of Metabolome and Transcriptome

All differentially expressed genes reported in our previous study with an adjusted *p*-value < 0.05 and |log2 fold change| > 1.5 [11] and differential metabolites were submitted to the KEGG pathways database (https://www.kegg.jp/kegg/pathway.html, accessed on 6 May 2025) to identify commonly enriched pathways. Subsequently, these differentially expressed elements were functionally annotated with a focus on their involvement in biological processes and signal transduction pathways based on the identified common pathways.

### 5.5. Detection of Cell Viability

5 × 10^3^ IPEC-J2 cells were cultured in 96-well plated for about 4 h, and then treated with *γ*GC (50–500 μM), DON (1 μg/mL) and co-treatment DON (1 μg/mL) and *γ*GC (50–500 μM). The CCK8 solution (Dojindo Laboratories, Kumamoto, Tokyo, Japan) was used to detected the cell viability. Tecan Infinite 200 microplate reader (Sunrise model, Tecan, Switzerland) was used for detecting absorbance at 450 nm, and calculate the cell viability of each group according to the instructions.

### 5.6. Flow Cytometry

For the analysis of ROS level, the treated cells were incubated with 10 μM of Dihydroethidium (DHE, KeyGEN BioTECH, Nanjing, China) at a temperature of 37 °C for 20 min, and then a flow cytometer (Excitation wavelengths: 488 nm, emission wavelengths: 525 nm, Beckman Coulter, Brea, CA, USA) was used to analysis the ROS level. For the analysis of apoptosis rates, the Annexin VPE/7-AAD (CA1020, Solarbio, Beijing, China) was used to staining the cells, and then a flow cytometer (Excitation wavelengths: 488 nm, emission wavelengths: 520 nm, Beckman Coulter, Brea, CA, USA) was used to analysis apoptosis rates. Each group consists of three replicates, and each sample was analyzed using 10,000 cells.

### 5.7. Western Blotting

The Western blotting procedure was conducted in accordance with the methodology outlined in our pervious study [37]. The simple steps are as follows: the RIPA buffer (Applygen, Beijing, China) was used to lyse the treated cells; subsequently, an equal amount of protein was separated by the SDS-PAGE method and transferred onto a PVDF membrane (Bio–Rad, Hercules, CA, USA). The membrane was blocked in 5% skim milk for 2 h, and then incubated with the primary antibody (BAX, BCL2, 1:5000) at 4 °C overnight.

Following three washes with TBST solution, the membrane was incubated at room temperature with HRP-conjugated secondary antibody (1:5000) for 2 h. The protein bands were visualized using ECL detection technology and quantitatively analyzed using ImageJ software (version 8.0.1). Anti-Bax (ER0907) and anti-Bcl-2 (SZ10-03) antibodies were obtained from HuaAn (Hangzhou, China). HSP90 (60318-1-lg) served as the loading control and was supplied by Proteintech Ltd. (Wuhan, China).

### 5.8. Statistical Analysis

All data are presented as mean ± standard derivation (SD). Student’s *t*-test or one-way ANOVA was performed to compare the statistical differences between different groups. The post hoc test was used to differentiate significance among different groups. A significance level of *p* < 0.05 was used to indicate statistical significance: * *p* < 0.05, ** *p* < 0.01.

## Figures and Tables

**Figure 1 toxins-17-00457-f001:**
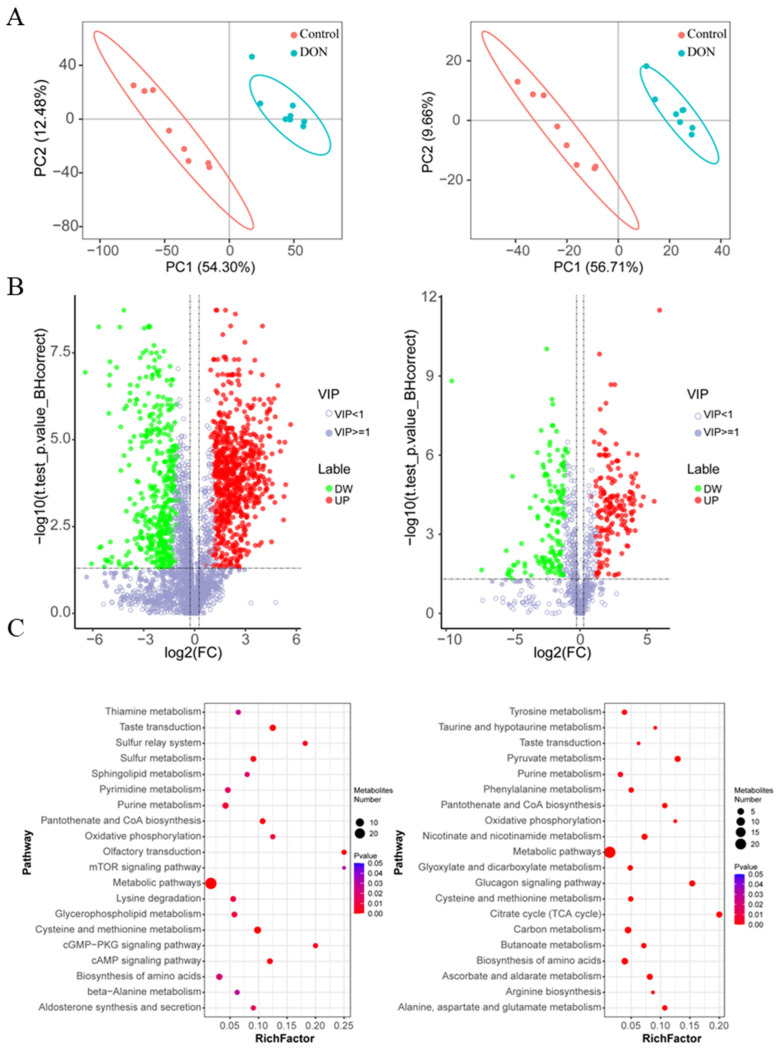
Metabolic analyses of IPEC-J2 cells upon DON exposure at the concertation of 1 µg/mL for 48 h. (**A**) Cluster analysis based on the metabolites by using the partial least-squares discrimination method. Positive (**left**)- and negative (**right**)-ion modes. (**B**) Volcano plots show differentially expressed metabolites in positive (**left**) and negative (**right**) ion modes. The hollow circles represent VIP ≤ 1, while the solid circles represent VIP > 1. The green solid circles indicate down regulated and the red solid circles indicate up regulated. (*n* = 8, |log2 fold change| > 1.2, and VIP > 1, q-value < 0.05). (**C**) The top 20 pathway based on the analysis of differential metabolites, positive (**left**)- and negative (**right**)-ion modes.

**Figure 2 toxins-17-00457-f002:**
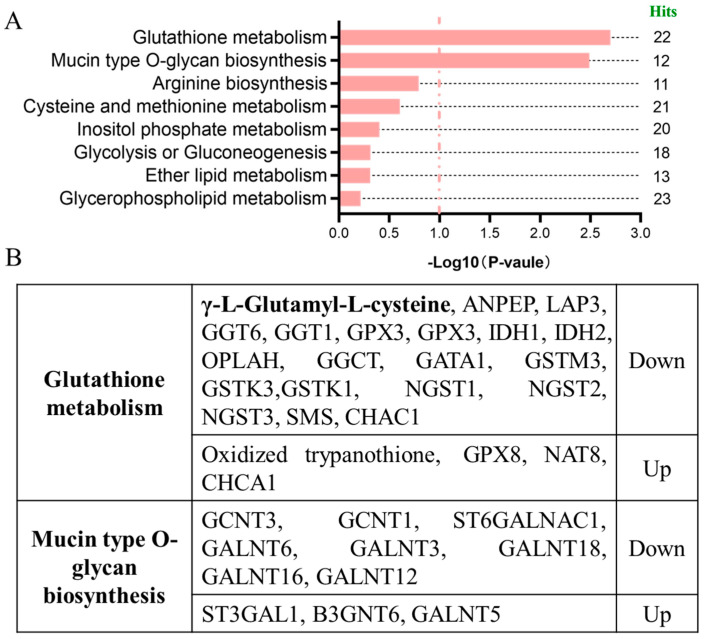
Integrated analysis of differential metabolites and differentially expressed genes of IPEC-J2 cells upon DON exposure. (**A**) Pathway analysis for differential metabolites and expressed genes of IPEC-J2 cells upon DON exposure. (**B**) Display of differential metabolites and differentially expressed genes in Glutathione metabolism and Mucin type O-glycan biosynthesis. γ-L-glutamyl-L-cysteine, *γ*GC. DON, Deoxynivalenol. The DON exposure at concertation of 1 µg/mL for 48 h.

**Figure 3 toxins-17-00457-f003:**
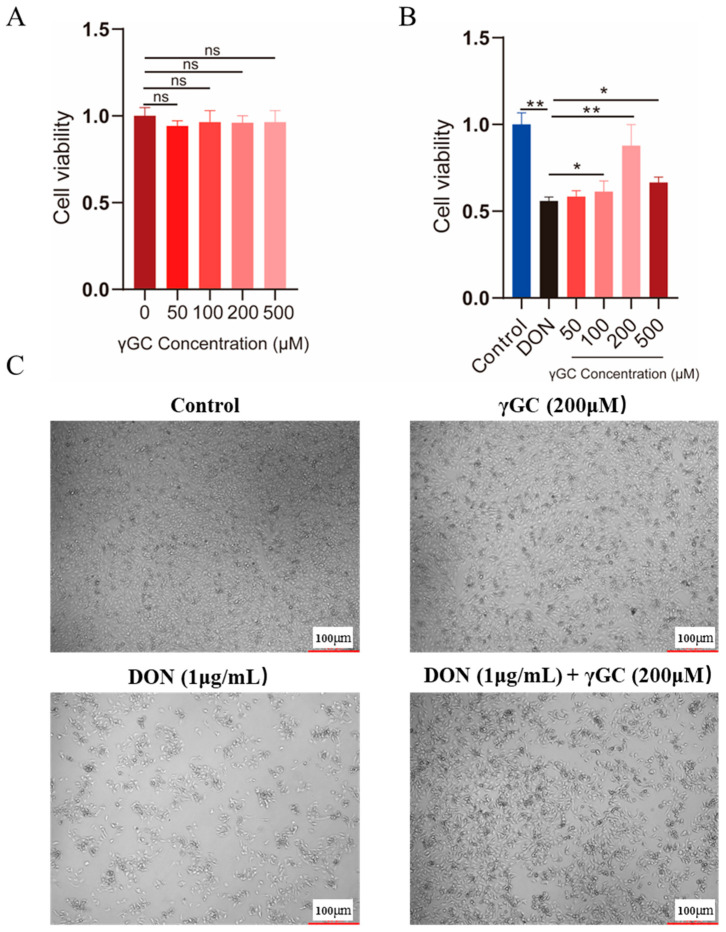
The protective effect of *γ*GC on the cytotoxicity induced by DON in the IPEC-J2 cells. (**A**) The effects of cell viability in different concentrations of *γ*GC. ns: no significantly. (**B**) The effects of cell viability in different concentrations of *γ*GC within the DON exposure. Control group: IPEC-J2 cells without DON and *γ*GC treatment; DON treated group: IPEC-J2 cells exposed to 1 μg/mL DON + *γ*GC group; 50–500 μM: cells treated with 1 μg/mL DON and various concentrations of *γ*GC. (*n* = 3). (**C**) The morphology of IPEC-J2 cells treated with Control, DON (1 μg/mL), *γ*GC (200μM) and DON (1 μg/mL) + *γ*GC (200 μM), Bar: 100 μm. Data are shown as mean ± standard derivation (SD). * *p* < 0.05 and ** *p* < 0.01.

**Figure 4 toxins-17-00457-f004:**
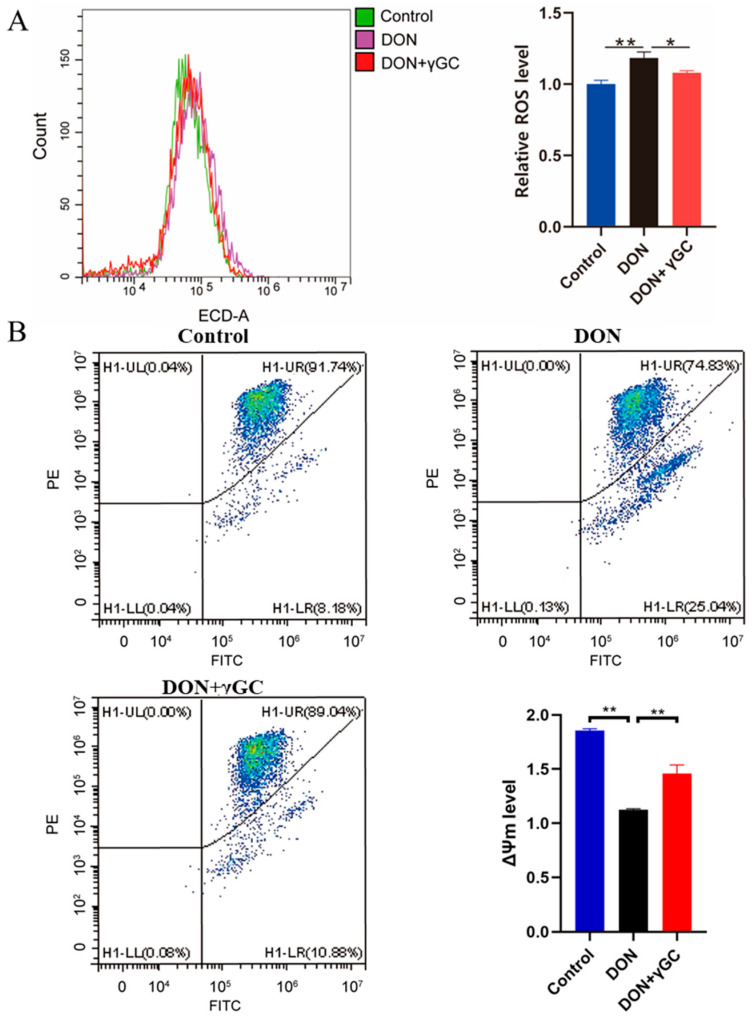
Effects of *γ*GC on ROS levels and mitochondrial function in IPEC-J2 cells upon DON exposure. (**A**) Fluorescence intensities of ROS in IPEC-J2 cells of the different groups. (**B**) Mitochondrial membrane potential levels (ΔΨm) of the different groups up on DON exposure and *γ*GC treatment. Control group: cells without DON and *γ*GC treatment; DON: cell treated with 1 μg/mL DON; DON + *γ*GC: cells exposed to 1 μg/mL DON and 200 μM *γ*GC (*n* = 3). Data are shown as mean ± SD. * *p* < 0.05 and ** *p* < 0.01.

**Figure 5 toxins-17-00457-f005:**
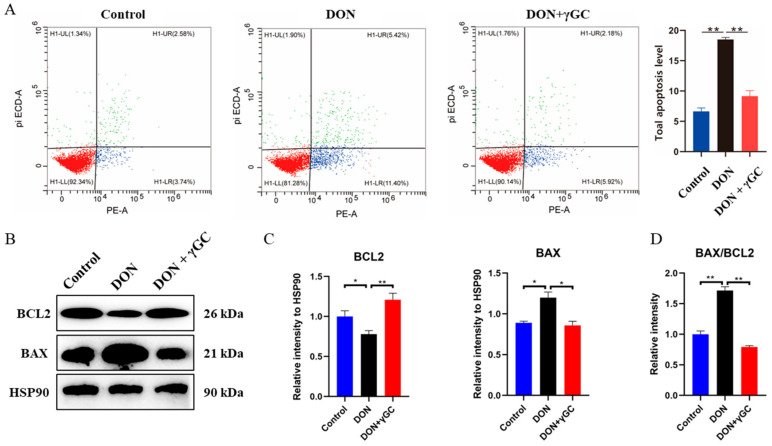
Effect of *γ*GC on IPEC-J2 cells apoptosis induced by DON. (**A**) Apoptosis of IPEC-J2 cells upon different treatment was evaluated by flow cytometry. (**B**) Western blotting analysis of the alterations in the protein BAX and BCL-2. (**C**) The relative intensity of BAX and BCL-2 upon different treatment (*n* = 3). (**D**) The ratio of BAX/BCL-2 in different treatments. Control: cells without DON and *γ*GC treatment; DON: cell treated with 1 μg/mL DON; DON + *γ*GC: cells exposed to 1 μg/mL DON and 200 μM *γ*GC. Data are shown as mean ± SD. * *p* < 0.05 and ** *p* < 0.01.

## Data Availability

The original contributions presented in this study are included in the article/Appendix A. Further inquiries can be directed to the corresponding author.

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
