# Peer review of "Integrative Analysis of Metabolome and Transcriptome Identifies the Role of γ-Glutamylcysteine in Mitigating Deoxynivalenol-Induced Toxicity"

_toxins, 2025, doi:10.3390/toxins17090457_

Round 1

Reviewer 1 Report

Comments and Suggestions for Authors

Authors:

General remarks: The authors presented a comprehensive paper, based on well designed and executed experiments. However, the manuscript requires major editorial revision, as the subchapter sequence is not consistent, and several minor errorous statement appear in the text. Main editorial comments re given below, and the many minor wordings minor requiring the attention are marked in colour in the pdf file.

Editorial comments

Line 34: should read: Fusarium species

Line 35-37: Delete the sentence: “During the storage of livestock …… significantly increasing the risk of contamination”. This statement is simply not correct as Fusarium are so-called field fungi and DON production occurs prior to harvest and not during storage. Moreover, the statement is not in line with the given reference (please add add a valid reference on DON feed contamination).  

Line 41-42:   delete: “circulatory shock and death” as this is not typically for DON (this is Fumonisin-related) and replace also reference 3 by more specific reference for DON.

Line 44: delete “liver” as DOM-1 is a product of bacterial metabolisms

Line 44- 45: the statement “in its original form” is not entirely correct, as DON is extensively glucuronidase prior to excretion.

Missing in the introduction: a clear working hypothesis (given later in the text, but should be added here.

Line 82: “GSH consumption-related diseases “a better wording is “GSH depleting…….  

Line 100-105: replace: these statements belong to the M&M section

Fig 1 and 2: Please complete the legend (needs to be self-explaining) by adducing the cell type, the DON concentrations, the duration of exposure to DON etc

Section 2.2. The authors jumped here into an “integrated analysis” while in 2.1. only the metabolic are presented. Please add at least a short explanatory section here.  Please avoid general statements (even when correct) on physiological mechanisms in the result section as for example lines 143-145 (need to be moved to discussion). This includes references, which are generally NOT given in the results section.  

Fig 3A and 3B: apparently different colour codes were use in the figure/ Please harmonize. Legend text: quite confusing, particularly line 177.

Line 180: Revisit the numbering and headings of the subchapters: now 2.2. again!

Figure 4 legend: DON concentrations are now suddenly given in µg/ml. What is correct?

Line 204: Chapter numbering and heading again to be corrected

Line 223: statement is not correct: DON (parent compound) binds to ribosomes but NOT after metabolization. DON changes the ribosome structure (not vice versa). Please rephrase these sentences.

Lines 223-230: Although correctly cited literature data, these explanations should be mentioned already in the introduction aa a part of the working hypothesis.

It is suggested to start the discussion with line 232.

Lines 257 – 287. The text “in summary” and “conclusions” are rather repetitive. Consider rephrasing and shorten (in summary is not essential)

Pleas revisited the reference list. Some of the articles are not closely linked to the subject of this manuscript.

Author Response

Response to Reviewer 1 Comments

Authors:

General remarks: The authors presented a comprehensive paper, based on well designed and executed experiments. However, the manuscript requires major editorial revision, as the subchapter sequence is not consistent, and several minor errorous statement appear in the text. Main editorial comments re given below, and the many minor wordings minor requiring the attention are marked in colour in the pdf file.

Reply: Thanks for your positive comments. We have prepared our revision according to the submission guidelines. In the revised manuscript, we have addressed the comments point-by-point.

Point 1: L. 34: should read: Fusarium species

Response 1: Thank you for your comment. These typos have been corrected. We decide to change the “Fusarium” into “Fusarium species”.

Point 2: L. 35-37: Delete the sentence: “During the storage of livestock …… significantly increasing the risk of contamination”. This statement is simply not correct as Fusarium are so-called field fungi and DON production occurs prior to harvest and not during storage. Moreover, the statement is not in line with the given reference (please add a valid reference on DON feed contamination).

Reply: Thank you for your comment about the DON production. We are agreeing with your suggestion, these typos have been corrected. As suggested, so we deleted the sentence “During the storage of livestock …… significantly increasing the risk of contamination”.

“Under unfavorable environmental conditions, such as high temperature and humidity, fungi may grow, reproduce, and release toxins during the cultivation or storage of grains.” [2]

  1. Eriksen GS, Knutsen HK, Sandvik M, Brantsæter A-L. Urinary deoxynivalenol as a biomarker of exposure in different age, life stage and dietary practice population groups. Environ Int. 2021, 157, 106804.
  2. Yue J, Guo D, Gao X, Wang J, Nepovimova E, Wu W, et al. Deoxynivalenol (Vomitoxin)-Induced Anorexia Is Induced by the Release of Intestinal Hormones in Mice. Toxins. 2021, 13.

Point 3: L. 41-42: delete: “circulatory shock and death” as this is not typically for DON (this is Fumonisin-related) and replace also reference 3 by more specific reference for DON.

Reply: Thanks for your comments. We have realized that our expression was inappropriate. The “circulatory shock and death” was deleted.

  1. Yue J, Guo D, Gao X, Wang J, Nepovimova E, Wu W, et al. Deoxynivalenol (Vomitoxin)-Induced Anorexia Is Induced by the Release of Intestinal Hormones in Mice. Toxins. 2021, 13.

Point 4: L. 44: delete “liver” as DOM-1 is a product of bacterial metabolisms

Reply: Thank you for your comment. These typos have been corrected.

Point 5: L. 44- 45: the statement “in its original form” is not entirely correct, as DON is extensively glucuronidase prior to excretion.

Reply: Thank you for your comment. Yes, we have understood your discussion regarding the metabolism of DON.

After pigs ingest DON, only a small amount is converted by the intestinal flora into the less toxic DOM-1 (De-epoxy-Deoxynivalenol) without epoxy groups and is excreted from the body in feces; The vast majority of DON is excreted in its original form, glucuronic acid-DON and glucuronic acid-DOM-1 through urine [5].

As suggested, the sentence was changed to “The vast majority of DON is excreted in its original form, glucuronic acid-DON and glucuronic acid-DOM-1 through urine”

Goossens J, Vandenbroucke V, Pasmans F, De Baere S, Devreese M, Osselaere A, et al. Influence of mycotoxins and a mycotoxin adsorbing agent on the oral bioavailability of commonly used antibiotics in pigs. Toxins. 2012, 4, 281-95.

Point 6: L. 82: “GSH consumption-related diseases “a better wording is “GSH depleting…….

Reply: Thank you for your comment. We decide to change “GSH consumption-related diseases” into “GSH depleting-related diseases”.

Point 7: L. 100-105: replace: these statements belong to the M&M section

Reply: Thank you for your suggestions. These typos have been corrected. These statements were replaced to the section of “Metabolites Extraction and LC-MS/MS Analysis”.

Point 8: Fig 1 and 2: Please complete the legend (needs to be self-explaining) by adducing the cell type, the DON concentrations, the duration of exposure to DON etc

Reply: Thank you for your valuable suggestion. Your input will contribute to enhancing the clarity of the article.

Figure 1. Metabolic analyses of IPEC-J2 cells upon DON exposure at the concertation of 1 µg/mL for 48 h. (A) Cluster analysis based on the me-tabolites in positive (left)- and negative (right)-ion modes using the partial least-squares discrimi-nation method. (B) Volcano plots of differential metabolites in positive (left)- and negative (right)-ion modes between the control and DON exposure groups. Red and green dots represent significantly upregulated and downregulated metabolites, respectively. (C) Pathway analysis of differential metabolites in positive (left)- and negative (right)-ion modes between the control and DON exposure groups. DON, Deoxynivalenol.

Figure 2. Integrated analysis of differential metabolites and differentially expressed genes of IPEC-J2 cells between the DON exposure and control groups. (A) Pathway analysis for differential metabolites and ex-pressed genes between the DON exposure and control groups. (B) Display of differential metabolites and differentially expressed genes in Glutathione metabolism and Mucin type O-glycan biosynthesis. γ-L-glutamyl-L-cysteine: γ-Glu-Cys, ?GC. DON, Deoxynivalenol. The DON exposure at the concertation of 1 µg/mL for 48 h.

Point 9: Section 2.2. The authors jumped here into an “integrated analysis” while in 2.1. only the metabolic are presented. Please add at least a short explanatory section here.  Please avoid general statements (even when correct) on physiological mechanisms in the result section as for example lines 143-145 (need to be moved to discussion). This includes references, which are generally NOT given in the results section.  

Reply: Thank you for your valuable comments and suggestion. This study systematically characterized metabolic profile changes after DON exposure. The identified differential metabolites are listed in Section 2.1. Section 2.2 then integrates transcriptomic data from previous studies to jointly analyse these metabolites and their associated differentially expressed genes, and simultaneously carried out pathway enrichment analysis.

Meanwhile, as suggested, the sentence of lines 143-145 was moved to discussion.

Point 10: Fig 3A and 3B: apparently different colour codes were use in the figure/ Please harmonize. Legend text: quite confusing, particularly line 177.

Reply: Thank you for your valuable comments and suggestion. As suggested, the colour of Fig 3A and 3B was harmonized. Meanwhile, the legend of Figure 3 was modified.

Figure 3. The protect effect of γGC on DON-induced cytotoxicity. (A) Effects of different concentrations of γGC on IPEC-J2 cells viability. (B) Effects of different concentrations of γGC on IPEC-J2 cells viability upon DON exposure. Control group: IPEC-J2 cells without DON and γGC treatment; DON treated group: IPEC-J2 cells exposed to 1μg/mL DON+ γGC group; 50-500μM: cells treated with 1μg/mL DON and various concentrations of γGC. (C) The morphology of IPEC-J2 cells treated with Control, DON (1μg/mL), γGC (200μM) and DON (1μg/mL) + γGC (200μM), Bar: 100μm. Data are shown as mean ± standard derivation (SD) (n=3). *p<0.05 and **p<0.01 compared with the control group.

Point 10: Line 180: Revisit the numbering and headings of the subchapters: now 2.2. again!

Reply: Thank you for your comment. These typos have been corrected.

“2.4 γGC Attenuated DON-induced Oxidative Stress in IPEC-J2 cells

2.5 γGC Attenuated DON-induced Apoptosis in IPEC-J2 cells”

Point 11: Figure 4 legend: DON concentrations are now suddenly given in µg/ml. What is correct?

Reply: Thank you for your comment. These typos have been corrected. In the study, the concentration of DON treatment was 1µg/ml and the concentration of γGC was given in μM.

Figure 4. Effects of γGC on ROS levels and mitochondrial function in IPEC-J2 cells upon DON exposure. (A) Fluorescence intensities of ROS in IPEC-J2 cells of the different groups. (B) Mitochondrial membrane potential levels (ΔΨm) of the different groups up on DON exposure and γGC treatment. Control group: cells without DON and γGC treatment; DON: cell treated with 1μg/mL DON; DON + γGC: cells exposed to 1μg/mL DON and 200 μM γGC. Data are shown as mean ± SD (n=3). *p<0.05 and **p<0.01.

Point 12: Line 204: Chapter numbering and heading again to be corrected

Reply: Thank you for your comment. These typos have been corrected.

“2.5 γGC Attenuated DON-induced Apoptosis in IPEC-J2 cells”

Point 13: Line 223: statement is not correct: DON (parent compound) binds to ribosomes but NOT after metabolization. DON changes the ribosome structure (not vice versa). Please rephrase these sentences.

Reply: Thank you for your comment. These typos have been corrected.

“Studies show that DON could bind ribosomes and chang its structure, disrupting their conformation and inhibiting peptidyl transferase activity on the 60S subunit, ultimately blocking the initiation, elongation, and termination of protein synthesis”

Point 14: Lines 223-230: Although correctly cited literature data, these explanations should be mentioned already in the introduction aa a part of the working hypothesis.

It is suggested to start the discussion with line 232.

Reply: Thank you for your comment. These typos have been corrected.

As suggested, the Lines 223-230 was deleted.

Point 15: Lines 257 – 287. The text “in summary” and “conclusions” are rather repetitive. Consider rephrasing and shorten (in summary is not essential)

Reply: Thank you for your comment. As suggested, the section of summary was deleted.

Point 16: Pleas revisited the reference list. Some of the articles are not closely linked to the subject of this manuscript.

Reply: Thank you for your comment. These types of reference list have been corrected.

Reviewer 2 Report

Comments and Suggestions for Authors

Dear Authors

I have carefully reviewed the manuscript entitled “Integrative Analysis of Metabolome and Transcriptome Identifies the Role of γ-glutamylcysteine in Mitigating Deoxynivalenol-induced Toxicity”. The study is well-designed, combining metabolomics and transcriptomics to provide novel insights into the toxicological mechanism of DON and the protective role of γGC. The integration of multi-omics data with functional assays strengthens the scientific value of this work. The experimental design is robust, the results are presented clearly, and the findings are relevant for both animal health and food safety. However, some minor concerns must be addressed to enhance the scientific clarity and presentation of the manuscript.

Moreover, there is a noticeable lack of updated and highly relevant references that could further strengthen the Introduction and Discussion sections.  I strongly recommend that the authors include the suggested references at appropriate points.

Comments:

Abstract

  • The wording in the abstract is imprecise and weakens the scientific impact of the study. A comprehensive revision is necessary to enhance clarity and scientific tone.
  • Please follow the abstract structure 1. Background, 2. Aim, 3. Methodology, 4. Results and 5. Conclusion.
  • Line 15-16 - Clarify whether “marked downregulation” refers to fold change or statistical significance.
  • Line 18-20- Rephrase “inhibit the apoptosis of IPEC-J2 cell induced by DON” to “inhibit DON-induced apoptosis in IPEC-J2 cells”.
  • Line 24-26- Replace “providing theoretical support” with “provides a theoretical basis” for conciseness.
  • Line 18-22- Include numerical values or percentage reductions for ROS/apoptosis.

Introduction

  • Line 38-40- Add more recent epidemiological data on DON contamination in feeds.
  • Line 47-50- Include tight junction damage with more recent molecular-level findings.
  • Lines 59-72 (discussion of metabolomics and transcriptomics in toxicology):
    The authors highlight the role of metabolomics and transcriptomics in uncovering toxicological mechanisms. I recommend citing Tang et al. (2023), who combined transcriptome and metabolome analyses to investigate mechanisms of intranasal insulin in vascular dementia, as it provides a strong methodological precedent for integrated multi-omics studies in toxin-related contexts.

Tang, L., Wang, Y., Gong, X., Xiang, J., Zhang, Y., Xiang, Q.,... Li, J. (2023). Integrated transcriptome and metabolome analysis to investigate the mechanism of intranasal insulin treatment in a rat model of vascular dementia. Frontiers in Pharmacology, 14, 1182803. doi: 10.3389/fphar.2023.1182803

  • To further strengthen the background on DON-induced metabolic dysregulation, I suggest referencing Wang et al. (2025), who reported DON-induced intestinal toxicity in zebrafish through disruption of amino acid metabolism and sphingolipid signaling pathways, which directly complements the theme of the present study.

Wang, Y., Wang, L., Du, Y., Yao, F., Zhao, M., Cai, C.,... Shao, S. (2025). Metabolomics study reveals DON-induced intestinal toxicity in adult zebrafish through disruption of amino acid metabolism and sphingolipid signaling pathway. Aquatic Toxicology, 282, 107324. doi: https://doi.org/10.1016/j.aquatox.2025.107324

  • Line 73-75- Provide more recent literature on GSH depletion in intestinal oxidative stress models.
  • Line 78-81- Clarify whether γGC has been tested in vivo for DON or only in other oxidative models.
  • Line 32-94- Add at least three updated references on DON toxicity mechanisms and antioxidant interventions.
  • Consider elaborating more on the research background, clearly stating the knowledge gap, the problem being addressed, and how the present study aims to provide the solution.
  • Conclude the introduction with a concise study aim statement.

Results

  • Include enrichment scores or adjusted p-values for pathways.
  • Add fold change value for γGC downregulation.
  • Quantify ROS reduction percentage with γGC treatment.
  • include fold changes for apoptosis reduction.

Discussion

  • Line 233-238- Discuss how your multi-omics findings compare with other DON metabolomics studies.
  • Lines 235-239 (multi-omics insights into biological processes):
    In discussing how metabolomics integrated with transcriptomics enhances mechanistic understanding, the authors should consider citing Han et al. (2024), who used Mendelian randomization to reveal causal links between metabolic factors and ovarian cancer risk. This supports the broader applicability of metabolic pathway insights in disease and toxin research.

Han, L., Xu, S., Zhou, D., Chen, R., Ding, Y., Zhang, M.,... Li, S. (2024). Unveiling the causal link between metabolic factors and ovarian cancer risk using Mendelian randomization analysis. Frontiers in Endocrinology, 15, 1401648. doi: 10.3389/fendo.2024.1401648

  • Line 255-257- Add references for γGC protective effects in other toxin models.
  • Line 260-263- Discuss whether γGC might act via pathways beyond GSH synthesis.
  • Line 264-266- Compare your findings with other antioxidant interventions for DON.
  • Lines 267-273 (novelty of detection and intervention strategies):
    When highlighting translational potential and future diagnostic/therapeutic strategies, the authors could also refer to Zhang et al. (2025), who developed a dual recombinase polymerase amplification with lateral flow immunoassay for pathogen detection. Although focused on microbes, it illustrates innovative biosensing approaches that align with the manuscript’s emphasis on developing interventions against DON toxicity.

Zhang, Y., Liu, X., Luo, J., Liu, H., Li, Y., Liu, J.,... Zeng, H. (2025). Dual recombinase polymerase amplification system combined with lateral flow immunoassay for simultaneous detection of Staphylococcus aureus and Vibrio parahaemolyticus. Journal of Pharmaceutical and Biomedical Analysis, 255, 116621. doi: https://doi.org/10.1016/j.jpba.2024.116621

  • Line 267-270- Suggest future in vivo validation experiments.

Conclusion

  • Line 274-287- Shorten conclusion to 4-5 sentences for better impact.
  • Line 276-278- Avoid repetition of results already stated in the discussion.
  • Line 278-282- Include potential industrial application of γGC in feed formulations.
  • Line 283-286- Rephrase “provides theoretical support” to “offers a mechanistic rationale.”

Materials and Methods

  • Justify why IPEC-J2 cells were chosen over other intestinal cell models.
  • Ensure correct formatting “CO2” should be “CO2”.
  • Line 289-291- Specify the passage number of IPEC-J2 cells to ensure reproducibility.
  • Line 294-296- Clarify whether γGC was co-treated or pre-treated in all experiments.
  • Line 299-301- State how confluence was verified before DON treatment (visual or automated).
  • Line 302-304- Provide LC-MS/MS instrument parameters (column type, flow rate, gradient).
  • Line 321-323- Clarify how metabolomics and transcriptomics data were normalized before integration.
  • Line 328-332- Include the manufacturer of the CCK-8 kit.
  • Line 334-340- Provide excitation/emission wavelengths for ROS detection.
  • Please mention the post hoc test to differentiate significance among treatments.

Author Response

Response to Reviewer 2 Comments

Dear Authors

I have carefully reviewed the manuscript entitled “Integrative Analysis of Metabolome and Transcriptome Identifies the Role of γ-glutamylcysteine in Mitigating Deoxynivalenol-induced Toxicity”. The study is well-designed, combining metabolomics and transcriptomics to provide novel insights into the toxicological mechanism of DON and the protective role of γGC. The integration of multi-omics data with functional assays strengthens the scientific value of this work. The experimental design is robust, the results are presented clearly, and the findings are relevant for both animal health and food safety. However, some minor concerns must be addressed to enhance the scientific clarity and presentation of the manuscript.

Moreover, there is a noticeable lack of updated and highly relevant references that could further strengthen the Introduction and Discussion sections.  I strongly recommend that the authors include the suggested references at appropriate points.

Reply: Thank you for your insightful comments and enthusiasms to our manuscript. We have prepared our revision according to the submission guidelines. In the revised manuscript, we have addressed the comments point-by-point.

Comments:

Abstract

  • The wording in the abstract is imprecise and weakens the scientific impact of the study. A comprehensive revision is necessary to enhance clarity and scientific tone.

Reply: Thank you for your comments. We appreciate your feedback and have thoroughly revised the abstract.

  • Please follow the abstract structure 1. Background, 2. Aim, 3. Methodology, 4. Results and 5. Conclusion.

Reply: Thank you for your comments. The abstract structure you provided is indeed very reasonable. We have made appropriate adjustments according to your suggestions.

Deoxynivalenol (DON), a prevalent environmental toxin produced by Fusarium fungi, frequently contaminates feed and food products. However, the critical metabolites and regulatory factors involved in DON toxicity remain poorly understood. Building upon our established DON-induced porcine intestinal epithelial cells (IPEC-J2) injury model. The study employed liquid chromatography-tandem mass spectrometry (LC-MS/MS) to conduct metabolomic analysis, and integrated analysis with transcriptomic data from DON-exposed IPEC-J2. Results identified 1,524 differentially expressed metabolites, and revealed significant enrichment in Glutathione metabolism and Mucin-type O-glycan biosyn-thesis pathways. Notably, γ-glutamylcysteine (γGC), the rate-limiting precursor for glutathione synthesis, showed significant reduction following DON exposure. To explore the biological function of γGC during DON exposure, this study found through exogenous supplementation experiments that γGC pretreatment could significantly alleviate the inhibition of IPEC-J2 cell viability, effectively reduce intracellular ROS accumulation and inhibit DON-induced apoptosis in IPEC-J2 cells. These results indicated that the severe oxidative stress induced by DON is closely related to the blockage of glutathione synthesis caused by the exhaustion of intracellular γGC, and revealed the application potential of γGC as an exogenous supplement in the prevention and treatment of DON exposure. In conclusion, this study provided global insights into metabolic and transcriptional changes as well as key metabolites and regulators un-derlying the cellular response to DON exposure, and provides a theoretical basis” for conciseness for the prevention and treatment of DON pollution.

  • Line 15-16 - Clarify whether “marked downregulation” refers to fold change or statistical significance.

Reply: Thank you for your comments.The intended meaning here is a significant reduction,To enhance the clarity of the description, we have implemented certain revisions in this section. “Showed significant reduction following DON exposure.”

  • Line 18-20- Rephrase “inhibit the apoptosis of IPEC-J2 cell induced by DON” to “inhibit DON-induced apoptosis in IPEC-J2 cells”.

Reply: Thank you for your comments. These typos have been corrected.

  • Line 24-26- Replace “providing theoretical support” with “provides a theoretical basis” for conciseness.

Reply: Thank you for your comments. These typos have been corrected.

  • Line 18-22- Include numerical values or percentage reductions for ROS/apoptosis.

Reply: Thank you for your comments.

Therefore, to further verify the protective effect of γGC in DON-induced oxidative stress, we detected the changes in intracellular reactive oxygen species (ROS) levels. The results indicated that DON exposure significantly induced the accumulation of ROS about 18.27%, while the supplementation of γGC significantly inhibited the in-crease of this ROS level about 8.74% (Figure 4A).

Flow cytometry was applied for detection, and the results showed that the level of DON-induced apoptosis was significantly reduced after γGC pretreatment during DON exposure (Control vs DON vs DON+γGC = 1.0+0.09 vs 2.78+0.05 vs 1.38+0.14) (Figure 5A).

Introduction

  • Line 38-40- Add more recent epidemiological data on DON contamination in feeds.

Reply: Thank you for your comments. As suggested, more epidemiological data on DON contamination in feeds were added in introduction.

“Over the 10-year period from 2008 to 2017, a total of 74,821 feed and feed raw material samples were collected globally, and the contamination rate of DON was found to be as high as 64% [1],however, a recent report on the detection of mycotoxins in feed samples collected in China between 2018 and 2020 revealed that the contamination rate of vomitoxin exceeded 96.4%, with average concentrations ranging from 458.0 to 1925.4 µg/kg [2].”

  1. Gruber-Dorninger C, Jenkins T, Schatzmayr G. Global Mycotoxin Occurrence in Feed: A Ten-Year Survey. Toxins (Basel). 2019, 11.

  1. hao L, Zhang L, Xu Z, Liu X, Chen L, Dai J, et al. Occurrence of Aflatoxin B1, deoxynivalenol and zearalenone in feeds in China during 2018-2020. Journal of Animal Science and Biotechnology. 2021, 12, 74.

  • Line 47-50- Include tight junction damage with more recent molecular-level findings.

Reply: Thank you for your comments.

“Previous studies have shown that DON exposure can disrupt the biological, physical and immune barriers of the intestinal tract, leading to a significant decrease in the ex-pression levels of mucins and tight junction proteins (such as claudin and occludin), thereby damaging the integrity of the intestinal barrier [8, 9] and making it easier to induce the invasion of other pathogenic microorganisms [8]. TLR2 ligand pre-treatment prevents DON-mediated intestinal barrier disruption by activating the PI3K/AKT signaling pathway [10].”

  • Lines 59-72 (discussion of metabolomics and transcriptomics in toxicology):
    The authors highlight the role of metabolomics and transcriptomics in uncovering toxicological mechanisms. I recommend citing Tang et al. (2023), who combined transcriptome and metabolome analyses to investigate mechanisms of intranasal insulin in vascular dementia, as it provides a strong methodological precedent for integrated multi-omics studies in toxin-related contexts.

Tang, L., Wang, Y., Gong, X., Xiang, J., Zhang, Y., Xiang, Q.,... Li, J. (2023). Integrated transcriptome and metabolome analysis to investigate the mechanism of intranasal insulin treatment in a rat model of vascular dementia. Frontiers in Pharmacology, 14, 1182803. doi: 10.3389/fphar.2023.1182803

Reply: Thank you for your comments. We have read this article, and indeed as you stated, it does provide a strong methodological precedent for integrated multi-omics studies in the context of toxins. So, we have cited the paper.

  1. Tang L, Wang Y, Gong X, Xiang J, Zhang Y, Xiang Q, et al. Integrated transcriptome and metabolome analysis to investigate the mechanism of intranasal insulin treatment in a rat model of vascular dementia. Front Pharmacol. 2023, 14, 1182803.

  • To further strengthen the background on DON-induced metabolic dysregulation, I suggest referencing Wang et al. (2025), who reported DON-induced intestinal toxicity in zebrafish through disruption of amino acid metabolism and sphingolipid signaling pathways, which directly complements the theme of the present study.

Wang, Y., Wang, L., Du, Y., Yao, F., Zhao, M., Cai, C.,... Shao, S. (2025). Metabolomics study reveals DON-induced intestinal toxicity in adult zebrafish through disruption of amino acid metabolism and sphingolipid signaling pathway. Aquatic Toxicology, 282, 107324. doi: https://doi.org/10.1016/j.aquatox.2025.107324

Reply: Thank you for your comments. As suggested, the Wang et al. (2025) was cited in introduction and discussion.

  1. Wang Y, Wang L, Du Y, Yao F, Zhao M, Cai C, et al. Metabolomics study reveals DON-induced intestinal toxicity in adult zebrafish through disruption of amino acid metabolism and sphingolipid signaling pathway. Aquat Toxicol. 2025, 282, 107324.

  • Line 73-75- Provide more recent literature on GSH depletion in intestinal oxidative stress models.

Reply: Thank you for your comments. As suggested, added more recent literature on GSH depletion in intestinal oxidative stress models.

Glutathione (GSH) is a key antioxidant within cells, maintaining REDOX homeo-stasis by scavenging free radicals and peroxides [18]. Its depletion is regarded as a key link in oxidative stress-induced intestinal inflammation and oxidative stress [19, 20].

  1. Moine L, Rivoira M, Díaz de Barboza G, Pérez A, Tolosa de Talamoni N. Glutathione depleting drugs, antioxidants and intestinal calcium absorption. World J Gastroenterol. 2018, 24, 4979-88.

  • Line 78-81- Clarify whether γGC has been tested in vivo for DON or only in other oxidative models.

Reply: Thank you for your comments.

“γGC, as a direct precursor of GSH synthesis, has good cellular uptake, can effectively bypass transmembrane transport barriers, and promote GSH synthesis. It is considered a potential target for regulating oxidative stress and improving GSH depleting-related diseases [22]. However, the changes of γGC in the oxidative stress of pig intestinal cells induced by DON and its potential regulatory mechanisms have not been reported yet.”

  • Line 32-94- Add at least three updated references on DON toxicity mechanisms and antioxidant interventions.

Reply: Thank you for your comments. As suggested, we added more updated references n DON toxicity mechanisms and antioxidant interventions in section of introduction.

“At the molecular level, previous study indicated that DON and its derivatives can damage cells by inhibiting protein synthesis by binding to the ribosome [3], disrupting proliferation, inhibiting the activation of various protein kinases and dysregulating gene expression [3-5]. Studies have shown that a diet containing 4 mg/kg of DON can significantly activate the MAPK and NF-κB signaling pathways in the jejunal and ileal tissues of weaned piglets [6]. The use of MAPK pathway inhibitors can significantly alleviate the cytotoxicity of DON on pig small intestinal epithelial cells (IPEC-J2) [7], inhibitors targeting NF-κB also alleviate the cell damage caused by DON [8].”

Tang M, Yuan D, Liao P. Berberine improves intestinal barrier function and reduces inflammation, immunosuppression, and oxidative stress by regulating the NF-κB/MAPK signaling pathway in deoxynivalenol-challenged piglets. Environmental Pollution (Barking, Essex: 1987). 2021, 289, 117865.

Zhang H, Deng X, Zhou C, Wu W, Zhang H. Deoxynivalenol Induces Inflammation in IPEC-J2 Cells by Activating P38 Mapk and Erk1/2. Toxins. 2020, 12.

Zhu M, Lu E-Q, Fang Y-X, Liu G-W, Cheng Y-J, Huang K, et al. Piceatannol Alleviates Deoxynivalenol-Induced Damage in Intestinal Epithelial Cells via Inhibition of the NF-κB Pathway. Molecules. 2024, 29.

  • Consider elaborating more on the research background, clearly stating the knowledge gap, the problem being addressed, and how the present study aims to provide the solution.
  • Conclude the introduction with a concise study aim statement.

Reply: Thank you for your comments. As suggested, we made more modification to clear our paper.

“In this study, in an attempt to address the aforementioned issues, we employed non-targeted liquid chromatography-tandem mass spectrometry (LC-MS) combined with transcriptome sequencing technology to systematically analyze the metabolic and transcriptional changes of IPEC-J2 intestinal epithelial cells after DON exposure. This discovery not only expands the understanding of the toxicological mechanism of DON, but also provides a theoretical basis for the future development of nutritional intervention strategies to prevent and control the toxicity of DON.”

Results

  • Include enrichment scores or adjusted p-values for pathways.

Reply: Thank you for your comments. The table about the adjusted p-values for pathways was as Table S3 upload with the revised manuscript.

  • Add fold change value for γGC downregulation.

Reply: Thank you for your comments. The P value (P<0.0237) and VIP (VIP=1.4317) was added in revised manuscript.

  • Quantify ROS reduction percentage with γGC treatment.

Reply: Thank you for your comments.

“The results indicated that DON exposure significantly induced the accumulation of ROS about 18.27%, while the supplementation of γGC significantly inhibited the increase of this ROS level about 8.74% (Figure 4A).”

  • include fold changes for apoptosis reduction.

Reply: Thank you for your comments.

“we examined the effect of DON exposure after γGC pretreatment of cells on the apop-tosis level of IPEC-J2 cells. Flow cytometry was applied for detection, and the results showed that the level of DON-induced apoptosis was significantly reduced after γGC pretreatment during DON exposure (Control vs DON vs DON+γGC = 1.0+0.09 vs 2.78+0.05 vs 1.38+0.14).”

Discussion

  • Line 233-238- Discuss how your multi-omics findings compare with other DON metabolomics studies.

Reply: Thank you for your comments. In introduction, we indicated that the advantage of the combination of metabolomics and RNA sequencing (RNA-seq).

“In recent years, the combination of metabolomics and RNA sequencing (RNA-seq) has been proven to effectively identify metabolite changes and related differentially ex-pressed genes (DEGs) caused by mycotoxin exposure or viral infection, providing a powerful tool for revealing toxicological mechanisms [22, 23]. Disturbances in the metabolic state of cells not only directly affect functions such as cell division and apoptosis, but can also be utilized by viruses or toxins to regulate the physiological processes of the host.”

“Our research first identified the changes in metabolites of intestinal epithelial cells after DON exposure. The differential metabolites were enriched in Thiamine metabolism, Taste transduction, Sulfur relay system, Sulfur metabolism, Sphingolipid metabolism and Tyrosine metabolism, Taurine and hypotaurine metabolism, Taste transduction, Pyruvate metabolism Signaling pathways such as Purine metabolism. In other DON metabolomics studies, it has been observed that the differential metabolites and enriched metabolic pathways resulting from DON toxicity exhibit significant variation. Such as, the toxic effects of DON on the intestinal tract of zebrafish are primarily mediated through alterations in 2-oxo-carboxylic acid metabolism, amino acid biosynthesis, carbon metabolism, and certain aspects of lipid metabolism  [9]. The reason for this might be due to the differences in the differential metabolites.”

  • Lines 235-239 (multi-omics insights into biological processes):
    In discussing how metabolomics integrated with transcriptomics enhances mechanistic understanding, the authors should consider citing Han et al. (2024), who used Mendelian randomization to reveal causal links between metabolic factors and ovarian cancer risk. This supports the broader applicability of metabolic pathway insights in disease and toxin research.

Han, L., Xu, S., Zhou, D., Chen, R., Ding, Y., Zhang, M.,... Li, S. (2024). Unveiling the causal link between metabolic factors and ovarian cancer risk using Mendelian randomization analysis. Frontiers in Endocrinology, 15, 1401648. doi: 10.3389/fendo.2024.1401648

Reply: Thank you for your comments. Yes, this article effectively elucidates the integration of metabolomics and transcriptomics analyses, clearly demonstrating the strong association between metabolic markers and the risk of ovarian cancer. However, the Mendelian randomization (MR) approach referenced in the article is not applicable to the current study. Instead, this research employs a combined analysis of differential metabolites and differentially expressed genes to identify key metabolites and significantly enriched pathways. Furthermore, the potential key metabolite γGC was identified in this study.

As suggested,the research was citing in our article.

[35] Han L, Xu S, Zhou D, Chen R, Ding Y, Zhang M, et al. Unveiling the causal link between metabolic factors and ovarian cancer risk using Mendelian randomization analysis. Front Endocrinol (Lausanne). 2024, 15, 1401648.

  • Line 255-257- Add references for γGC protective effects in other toxin models.

Reply: Thank you for your comments. We fully acknowledge the limitations in our current study regarding the functional exploration of γ-GC and its protective mechanisms in toxin models.

“Early experimental studies have demonstrated that intraperitoneal administration of γ-glutamylcysteine (γ-GC) can effectively restore glutathione levels depleted in various organs [10], and exerts hepatoprotective effects against liver injury induced by iron overload [11]. Furthermore, γ-GC has been shown to ameliorate oxidative damage in cultured neurons and astrocytes in vitro, while increasing cerebral glutathione content in vivo [12]. In addition, γ-GC confers protection against toxicity induced by cecal ligation and puncture (CLP) or lipopolysaccharide (LPS) in murine models, with its anti-inflammatory mechanisms involving the elevation of intracellular glutathione (GSH) levels and the reduction of reactive oxygen species (ROS) accumulation  [13]. These findings collectively suggest that γ-GC may exert protective effects against deoxynivalenol (DON)-induced oxidative stress and inflammatory responses in intestinal cells. However, to date, no studies have specifically investigated the protective mechanisms of γ-GC in pathological damage caused by related mycotoxins.”

Anderson ME, Meister A. Transport and direct utilization of gamma-glutamylcyst(e)ine for glutathione synthesis. P Natl Acad Sci Usa. 1983, 80, 707-11.

Salama SA, Al-Harbi MS, Abdel-Bakky MS, Omar HA. Glutamyl cysteine dipeptide suppresses ferritin expression and alleviates liver injury in iron-overload rat model. Biochimie. 2015, 115, 203-11.

Le TM, Jiang H, Cunningham GR, Magarik JA, Barge WS, Cato MC, et al. γ-Glutamylcysteine ameliorates oxidative injury in neurons and astrocytes in vitro and increases brain glutathione in vivo. Neurotoxicology. 2011, 32, 518-25.

Yang Y, Li L, Hang Q, Fang Y, Dong X, Cao P, et al. γ-glutamylcysteine exhibits anti-inflammatory effects by increasing cellular glutathione level. Redox Biol. 2019, 20, 157-66.

  • Line 260-263- Discuss whether γGC might act via pathways beyond GSH synthesis.

Reply: Thank you for your comments. In this study, it was clearly demonstrated that ?GC supplementation could inhibit the accumulation of ROS and apoptosis induced by DON exposure by increasing the antioxidant capacity of cells, this result is consistent with the excellent antioxidant effect of γGC. However, as mentioned in the above issues, it also has a significant anti-inflammatory effect. Moreover, our previous research indicated that DON exposure can significantly cause inflammatory responses in intestinal epithelial cells [14]. Therefore, we speculate that γGC might play a role in DON-induced intestinal epithelial cell damage through its anti-inflammatory properties.

As suggested, we added discussion about the other potential pathway.

“Moreover, our previous research indicated that DON exposure can significantly cause inflammatory responses in intestinal epithelial cells [14]. Therefore, we speculate that γGC might play a role in DON-induced intestinal epithelial cell damage through its anti-inflammatory properties.”

Zhu X, Wu J, Chen X, Shi D, Hui P, Wang H, et al. DNA ligase III mediates deoxynivalenol exposure-induced DNA damage in intestinal epithelial cells by regulating oxidative stress and interaction with PCNA. International Journal of Biological Macromolecules. 2024, 282, 137137.

  • Line 264-266- Compare your findings with other antioxidant interventions for DON.

Reply: Thank you for your comments. Yes, γGC demonstrates notable antioxidant properties. In this study, it was further confirmed that γGC can significantly alleviate the oxidative stress response in porcine intestinal epithelial cells induced by DON. Previous studies have shown that melatonin, owing to its potent antioxidant capacity, can mitigate DON-induced oxidative stress and markedly inhibit autophagy in intestinal epithelial cells [15]. These findings suggest that various compounds may modulate DON-induced oxidative stress through diverse mechanistic pathways.

“Previous studies have shown that melatonin, owing to its potent antioxidant capacity, can mitigate DON-induced oxidative stress and markedly inhibit autophagy in intestinal epithelial cells [15]. These findings suggest that various compounds may modulate DON-induced oxidative stress through diverse mechanistic pathways.”

Xu Y, Xie Y, Wu Z, Wang H, Chen Z, Wang J, et al. Protective effects of melatonin on deoxynivalenol-induced oxidative stress and autophagy in IPEC-J2 cells. Food and Chemical Toxicology: an International Journal Published For the British Industrial Biological Research Association. 2023, 177, 113803.

  • Lines 267-273 (novelty of detection and intervention strategies):
    When highlighting translational potential and future diagnostic/therapeutic strategies, the authors could also refer to Zhang et al. (2025), who developed a dual recombinase polymerase amplification with lateral flow immunoassay for pathogen detection. Although focused on microbes, it illustrates innovative biosensing approaches that align with the manuscript’s emphasis on developing interventions against DON toxicity.

Zhang, Y., Liu, X., Luo, J., Liu, H., Li, Y., Liu, J.,... Zeng, H. (2025). Dual recombinase polymerase amplification system combined with lateral flow immunoassay for simultaneous detection of Staphylococcus aureus and Vibrio parahaemolyticus. Journal of Pharmaceutical and Biomedical Analysis, 255, 116621. doi: https://doi.org/10.1016/j.jpba.2024.116621

Reply: Thank you for your comments. Yes, the article developed a dual recombinase polymerase amplification with lateral flow immunoassay for pathogen detection, and it illustrates innovative biosensing approaches. Future research will utilize the concepts outlined in this article to develop technical methodologies for the detection of DON content and its associated toxicity.

  • Line 267-270- Suggest future in vivo validation experiments.

Reply: Thank you for your comments. As suggested, the future study include the vivo validation experiments.

Conclusion

  • Line 274-287- Shorten conclusion to 4-5 sentences for better impact.
  • Line 276-278- Avoid repetition of results already stated in the discussion.
  • Line 278-282- Include potential industrial application of γGC in feed formulations.
  • Line 283-286- Rephrase “provides theoretical support” to “offers a mechanistic rationale.”

Reply: Thank you for your comments. These typos have been corrected. Based on the above suggestions, we made appropriate revisions to the conclusion section.

“In conclusion, this study confirmed that the metabolic characteristics of intestinal epithelial cells were significantly reshaped under DON exposure conditions. These alterations are closely related to the cytotoxicity induced by DON. In particular, the level of γGC was significantly downregulated after DON exposure, which was highly correlated with the increase in intracellular oxidative stress levels. Further functional verification experiments demonstrated that exogenous supplementation of γGC could effectively alleviate the oxidative stress response induced by DON. The study not only deepens the understanding of the toxicological mechanism of DON, especially its regulation of oxidative stress and metabolic homeostasis, but also offers a mechanistic rationale and experimental basis for the application of γGC as a potential nutritional intervention in the prevention and mitigation of DON pollution-related toxicity.”

Materials and Methods

  • Justify why IPEC-J2 cells were chosen over other intestinal cell models.

Reply: Thank you for your comments. IPEC-J2 cells are derived from neonatal piglet jejunum, making them highly relevant for studies on swine-specific pathogens (e.g., PCV2, PEDV) or agricultural toxins (e.g., DON, common in swine feed). Therefore, our term focused on investigating the pathological and molecular mechanisms of intestinal damage caused by PEDV, TGEV, PDCOV, DON and ZEA using IPEC-J2 cells as the experimental subjects.

  • Ensure correct formatting “CO2” should be “CO2”.

Reply: Thank you for your suggestions. These typos have been corrected.

  • Line 289-291- Specify the passage number of IPEC-J2 cells to ensure reproducibility.

Reply: Thank you for your suggestions.

“IPEC-J2 cells were seeded in six-well plates at a density of 5 × 105 cells /mL”

  • Line 294-296- Clarify whether γGC was co-treated or pre-treated in all experiments.

Reply: Thank you for your suggestions. These typos have been corrected, the γGC was co-treated in all experiments.

“IPEC-J2 cells were cultured in 96-well plated and treated as described above. For the co-treatment of γGC, cells were treated with concentrations of γGC (50-500 μM) and DON exposure. The CCK8 solution was dissolved in the medium and then added to each well. Then, cells were incubated at 37℃ for 3h. The absorbance at 450 nm was measured using a Tecan Infinite 200 microplate reader (Sunrise model, manufactured by Tecan in Switzerland).”

  • Line 299-301- State how confluence was verified before DON treatment (visual or automated).

Reply: Thank you for your suggestions. Regarding the issue of cell confluence, we made the judgment based on our experience with the procedure and observations.

“When the cell fusion degree is 60%-70% based on our experience with the procedure and observations”

  • Line 302-304- Provide LC-MS/MS instrument parameters (column type, flow rate, gradient).

Reply: Thank you for your suggestions. The LC-MS/MS instrument was High-resolution mass spectrometer Q Exactive made by Thermo Fisher Scientific, USA.

Based on the above suggestions, we made appropriate revisions.

“the LC-MS/MS (High-resolution mass spectrometer Q Exactive, Thermo Fisher Scientific, USA) were performed by BIG (Shenzhen, China)”

  • Line 321-323- Clarify how metabolomics and transcriptomics data were normalized before integration.

Reply: Thank you for your suggestions. These typos have been corrected.

“All differentially expressed genes reported in our previous study with an adjusted P-value<0.05 and |log2 fold change| > 1.5”

About the transcriptomics data was Clarify in the section of 5.3 Metabolomics Analysis.

  • Line 328-332- Include the manufacturer of the CCK-8 kit.

Reply: Thank you for your suggestions. These typos have been corrected.

. The CCK8 solution (Dojindo Laboratories, Kumamoto, Tokyo, Japan) was dissolved in the medium and then added to each well. Then, cells were incubated at 37℃ for 3h.

  • Line 334-340- Provide excitation/emission wavelengths for ROS detection.

Reply: Thank you for your suggestions. These typos have been corrected.

“Excitation wavelengths:488nm, emission wavelengths: 525nm.”

  • Please mention the post hoc test to differentiate significance among treatments.

Reply: Thank you for your comments. These typos have been corrected.

“The post-hoc tests was used to differentiate significance among different groups”

Gruber-Dorninger C, Jenkins T, Schatzmayr G. Global Mycotoxin Occurrence in Feed: A Ten-Year Survey. Toxins. 2019, 11.

Zhao L, Zhang L, Xu Z, Liu X, Chen L, Dai J, et al. Occurrence of Aflatoxin B1, deoxynivalenol and zearalenone in feeds in China during 2018-2020. Journal of Animal Science and Biotechnology. 2021, 12, 74.

Pestka JJ. Deoxynivalenol: mechanisms of action, human exposure, and toxicological relevance. Archives of Toxicology. 2010, 84, 663-79.

Wang H, Zong Q, Wang S, Zhao C, Wu S, Bao W. Genome-Wide DNA Methylome and Transcriptome Analysis of Porcine Intestinal Epithelial Cells upon Deoxynivalenol Exposure. J Agric Food Chem. 2019, 67, 6423-31.

Fan H, Wang S, Wang H, Sun M, Wu S, Bao W. Melatonin Ameliorates the Toxicity Induced by Deoxynivalenol in Murine Ovary Granulosa Cells by Antioxidative and Anti-Inflammatory Effects. Antioxidants (Basel). 2021, 10, 1045.

Tang M, Yuan D, Liao P. Berberine improves intestinal barrier function and reduces inflammation, immunosuppression, and oxidative stress by regulating the NF-κB/MAPK signaling pathway in deoxynivalenol-challenged piglets. Environmental Pollution (Barking, Essex : 1987). 2021, 289, 117865.

Zhang H, Deng X, Zhou C, Wu W, Zhang H. Deoxynivalenol Induces Inflammation in IPEC-J2 Cells by Activating P38 Mapk And Erk1/2. Toxins. 2020, 12.

Zhu M, Lu E-Q, Fang Y-X, Liu G-W, Cheng Y-J, Huang K, et al. Piceatannol Alleviates Deoxynivalenol-Induced Damage in Intestinal Epithelial Cells via Inhibition of the NF-κB Pathway. Molecules. 2024, 29.

Wang Y, Wang L, Du Y, Yao F, Zhao M, Cai C, et al. Metabolomics study reveals DON-induced intestinal toxicity in adult zebrafish through disruption of amino acid metabolism and sphingolipid signaling pathway. Aquat Toxicol. 2025, 282, 107324.

Anderson ME, Meister A. Transport and direct utilization of gamma-glutamylcyst(e)ine for glutathione synthesis. P Natl Acad Sci Usa. 1983, 80, 707-11.

Salama SA, Al-Harbi MS, Abdel-Bakky MS, Omar HA. Glutamyl cysteine dipeptide suppresses ferritin expression and alleviates liver injury in iron-overload rat model. Biochimie. 2015, 115, 203-11.

Le TM, Jiang H, Cunningham GR, Magarik JA, Barge WS, Cato MC, et al. γ-Glutamylcysteine ameliorates oxidative injury in neurons and astrocytes in vitro and increases brain glutathione in vivo. Neurotoxicology. 2011, 32, 518-25.

Yang Y, Li L, Hang Q, Fang Y, Dong X, Cao P, et al. γ-glutamylcysteine exhibits anti-inflammatory effects by increasing cellular glutathione level. Redox Biol. 2019, 20, 157-66.

Zhu X, Wu J, Chen X, Shi D, Hui P, Wang H, et al. DNA ligase III mediates deoxynivalenol exposure-induced DNA damage in intestinal epithelial cells by regulating oxidative stress and interaction with PCNA. International Journal of Biological Macromolecules. 2024, 282, 137137.

Xu Y, Xie Y, Wu Z, Wang H, Chen Z, Wang J, et al. Protective effects of melatonin on deoxynivalenol-induced oxidative stress and autophagy in IPEC-J2 cells. Food and Chemical Toxicology : an International Journal Published For the British Industrial Biological Research Association. 2023, 177, 113803.

Reviewer 3 Report

Comments and Suggestions for Authors

An interesting study related to DON-induced oxidative stress, which is linked to the inhibition of glutathione synthesis caused by intracellular γGC depletion, which revealed the potential application of γGC as an exogenous supplement in the prevention and treatment of DON exposure. However, in Materials and Methods at 5.3. Metabolomics Analysis, related to LC-MS/MS method, data concerning  the limit of detection and recovery rate should be added, especially since a bibliographic reference (25) is given in line 310 that is not found in the list of bibliographic titles, so a correction of Bibliography is required ( the entire list contains only 24 references). On the other hand, the Discussion chapter can be developed considering the results obtained, and also more bibliographic sources should be studied and added to this chapter.

Author Response

Response to Reviewer 3 Comments

Reviewer 3:

An interesting study related to DON-induced oxidative stress, which is linked to the inhibition of glutathione synthesis caused by intracellular γGC depletion, which revealed the potential application of γGC as an exogenous supplement in the prevention and treatment of DON exposure. However, in Materials and Methods at 5.3. Metabolomics Analysis, related to LC-MS/MS method, data concerning the limit of detection and recovery rate should be added, especially since a bibliographic reference (25) is given in line 310 that is not found in the list of bibliographic titles, so a correction of Bibliography is required (the entire list contains only 24 references). On the other hand, the Discussion chapter can be developed considering the results obtained, and also more bibliographic sources should be studied and added to this chapter.

Reply:

Thanks for your positive comments. We are glad that you are interested in our findings. we have prepared our revision according to the submission guidelines.

As your suggested, we have made appropriate adjustments according to your suggestion in Materials and Methods at 5.3, bibliographic reference, Discussion chapter.

Abstract

Deoxynivalenol (DON), a prevalent environmental toxin produced by Fusarium fungi, frequently contaminates feed and food products. However, the critical metabolites and regulatory factors involved in DON toxicity remain poorly understood. Building upon our established DON-induced porcine intestinal epithelial cells (IPEC-J2) injury model. The study employed liquid chromatography-tandem mass spectrometry (LC-MS/MS) to conduct metabolomic analysis, and integrated analysis with transcriptomic data from DON-exposed IPEC-J2. Results identified 1,524 differentially expressed metabolites, and revealed significant enrichment in Glutathione metabolism and Mucin-type O-glycan biosyn-thesis pathways. Notably, γ-glutamylcysteine (γGC), the rate-limiting precursor for glutathione synthesis, showed significant reduction following DON exposure. To explore the biological function of γGC during DON exposure, this study found through exogenous supplementation experiments that γGC pretreatment could significantly alleviate the inhibition of IPEC-J2 cell viability, effectively reduce intracellular ROS accumulation and inhibit DON-induced apoptosis in IPEC-J2 cells. These results indicated that the severe oxidative stress induced by DON is closely related to the blockage of glutathione synthesis caused by the exhaustion of intracellular γGC, and revealed the application potential of γGC as an exogenous supplement in the prevention and treatment of DON exposure. In conclusion, this study provided global insights into metabolic and transcriptional changes as well as key metabolites and regulators underlying the cellular response to DON exposure, and provides a theoretical basis” for conciseness for the prevention and treatment of DON pollution.

  1. Discussion

Studies show that after DON is metabolized, it binds to ribosomes by changing its structure, disrupting their conformation and inhibiting peptidyl transferase activity on the 60S subunit, ultimately blocking the initiation, elongation, and termination of protein synthesis [31]. Meanwhile, this "ribosomal toxic stress response" can rapidly activate double-stranded RNA-activated protein kinase R, which in turn activates the phosphorylation of the endogenous mitogen-activated protein kinase (MAPK) signaling pathway [32], thereby influencing downstream cellular physiological processes, including inducing apoptosis, hindering the cell cycle process, triggering inflammatory responses and oxidative stress [32]. Metabolomics analysis can identify metabolites that can regulate different biological processes and further shape the phenotypes of cells or organisms [33]. In this study, we conducted a metabolomics study on intestinal cells exposed to DON and detected metabolites with significant changes. Emerging evidence indicates that active metabolites can affect various aspects of the omics landscape, and comprehensive analysis with other omics data can provide mechanistic insights into related biological processes [34, 35].Therefore, our comprehensive analysis of metabolic and transcriptional data revealed the main pathways affected by DON exposure and provided a systematic diagram of the biological modules that regulate cells' toxic responses to DON.

Our research first identified the changes in metabolites of intestinal epithelial cells after DON exposure. The differential metabolites were enriched in Thiamine metabolism, Taste transduction, Sulfur relay system, Sulfur metabolism, Sphingolipid metabolism and Tyrosine metabolism, Taurine and hypotaurine metabolism, Taste transduction, Pyruvate metabolism Signaling pathways such as Purine metabolism. In other DON metabolomics studies, it has been observed that the differential metabolites and enriched metabolic pathways resulting from DON toxicity exhibit significant variation. Such as, the toxic effects of DON on the intestinal tract of zebrafish are primarily mediated through alterations in 2-oxo-carboxylic acid metabolism, amino acid biosynthesis, carbon metabolism, and certain aspects of lipid metabolism  [9]. The reason for this might be due to the differences in the differential metabolites. In our study, we found that the differential metabolite ?GC is a direct precursor of GSH and is significantly downregulated after DON exposure. It seems reasonable to speculate that exogenous ?GC enters cells and is enzymatically transformed by GSS, thereby increasing intracellular GSH concentration to counteract DON-induced oxidative stress. Studies have shown that GSH depletion can lead to the accumulation of ROS and exacerbate cell damage [36]. As an indispensable antioxidant, glutathione maintains the REDOX balance within cells by eliminating free radicals and peroxides [24]. Can directly providing GSH here alleviate the cell damage induced by DON? The study found that the uptake of extracellular GSH is restricted by ? -glutamyl transpeptidase, and its de noo synthesis is also limited by two key rate-limiting steps: Glutamic acid and cysteine are converted to ?GC through the following action of ?-glutamylcysteine ligase, and then from ?GC to glycine, promoted by glutathione synthase [27]. ?GC is a direct precursor of GSH, which can bypass the thermodynamic barriers related to transmembrane transport and is easily absorbed by various cells to promote GSH synthesis. Therefore, ?GC is regarded as a target for treating some diseases [37]. Early experimental studies have demonstrated that intraperitoneal administration of γ-glutamylcysteine (γ-GC) can effectively restore glutathione levels depleted in various organs [38], and exerts hepatoprotective effects against liver injury induced by iron overload [39]. Furthermore, γ-GC has been shown to ameliorate oxidative damage in cultured neurons and astrocytes in vitro, while increasing cerebral glutathione content in vivo [40]. In addition, γ-GC confers protection against toxicity induced by cecal ligation and puncture (CLP) or lipopolysaccharide (LPS) in murine models, with its anti-inflammatory mechanisms involving the elevation of intracellular glutathione (GSH) levels and the reduction of reactive oxygen species (ROS) accumulation  [27]. These findings collectively suggest that γ-GC may exert protective effects against deoxynivalenol (DON)-induced oxidative stress and inflammatory responses in intestinal cells. However, to date, no studies have specifically investigated the protective mechanisms of γ-GC in pathological damage caused by related mycotoxins.

In this study, it was also found that ?GC has no significant toxic effect on cell viability, and γGC significantly alleviates the reduction of cell activity induced by DON within a specific concentration range. And it shows a dose-dependent trend. This result preliminarily indicates that ?GC plays a role in resisting DON exposure-induced cell damage in intestinal cells. The detection of ROS levels and apoptosis levels further confirmed that ?GC supplementation could inhibit the accumulation of ROS and apoptosis induced by DON exposure by increasing the antioxidant capacity of cells, and to a limited extent alleviate the intestinal cell damage induced by DON exposure. This result is consistent with the function of the reported ?GC. Moreover, our previous research indicated that DON exposure can significantly cause inflammatory responses in intestinal epithelial cells [41]. Therefore, we speculate that γGC might play a role in DON-induced intestinal epithelial cell damage through its anti-inflammatory properties. Previous studies have shown that melatonin, owing to its potent antioxidant capacity, can mitigate DON-induced oxidative stress and markedly inhibit autophagy in intestinal epithelial cells [30]. These findings suggest that various compounds may modulate DON-induced oxidative stress through diverse mechanistic pathways.

  1. Conclusions

In conclusion, this study confirmed that the metabolic characteristics of intestinal epithelial cells were significantly reshaped under DON exposure conditions. These alterations are closely related to the cytotoxicity induced by DON. In particular, the level of γGC was significantly downregulated after DON exposure, which was highly correlated with the increase in intracellular oxidative stress levels. Further functional verification experiments demonstrated that exogenous supplementation of γGC could effectively alleviate the oxidative stress response induced by DON. The study not only deepens the understanding of the toxicological mechanism of DON, especially its regulation of oxidative stress and metabolic homeostasis, but also offers a mechanistic rationale and experimental basis for the application of γGC as a potential nutritional intervention in the prevention and mitigation of DON pollution-related toxicity.

  1. Materials and Methods

5.1 Cell Culture and DON Treatment

The IPEC-J2 cells line was used to build the model of DON exposure, and the cells was incubated with Dulbecco’s Modified Eagle Medium (DMEM) containing 10% fetal bovine serum (FBS, Gibco, Grand Island, NY, USA), in 5% CO2 at 37°C. IPEC-J2 cells were seeded in six-well plates at a density of 5 × 105 cells /mL. According to our previous, the model was treated with 1 µg/mL DON (C15H20O6; Sigma-Aldrich, St. Louis, MO, USA) for 48 h [11]. DON was dissolved in DMSO (initial concentration of 2 mg/ml) and diluted in culture medium to get a final concentration of 1 µg/ml. For γGC (γ-Glutamylcysteine, MCE, HY-113402, Shanghai, China) supplementation, IPEC-J2 cells were treated with 200 -500μM γGC during DON exposure.

5.2 Metabolites Extraction and LC-MS/MS Analysis

To explore the metabolite changes involved in the response of intestinal epithelial cells to DON exposure, this study was based on the previously established DON-induced injury model of IPEC-J2 cells [11]. Subsequently, LC-MS/MS technology was used to conduct non-targeted metabolomics analyses on the samples of the DON treatment group and the control group in both positive ion and negative ion modes. IPEC-J2 cells were seeded in six-well plates at a density of 5 × 105 cells /mL and cultured in an incubator at 37 ° C and 5% CO2. When the cell fusion degree is 60%-70% based on our experience with the procedure and observations, add 1 μg/mL of DON to the cells and culture for 48 hours. Eight biological replicates in DON treated and Control IPEC-J2 cells were set up for metabolomics detection. The sample processing method referred to the article we published previously [21] and the LC-MS/MS (High-resolution mass spectrometer Q Exactive, Thermo Fisher Scientific, USA) were performed by BIG (Shenzhen, China).

5.3 Metabolomics Analysis

The Compound Discoverer 3.1 (Thermo Fisher Scientific, USA) software was used to process the LC-MS/MS data., mainly including peak extraction, peak alignment and compound identification. Data preprocessing, statistical analysis, metabolite classification annotation and functional annotation was carried out using the self-developed metabolomics R software package metaX [41] and the metabolomics information analysis process. The original data of multiple variables is dimensionally reduced through PCA (Principal Component Analysis) to analyze the grouping, trends (similarity and difference within and between sample groups), and outliers (whether there are abnormal samples) of the observed variables in this dataset. Peak is matched with mzCloud, mzVault and Masslist databases (Thermo Fisher Scientific, Waltham) to obtain accurate and relative quantitative results. And according to the KEGG (https://www.genome.jp/kegg/pathway.html) for metabolites. Differentially expressed metabolites were identified with fold change >1.2 or <0.83, and VIP > 1, q-value < 0.05 between the two groups.

5.4 Integrated Analysis of Metabolome and Transcriptome

All differentially expressed genes reported in our previous study with an adjusted P-value<0.05 and |log2 fold change| > 1.5 [11] and differential metabolites were submitted to the KEGG pathways database (https://www.kegg.jp/kegg/pathway.html) to identify commonly enriched pathways. Subsequently, these differentially expressed elements were functionally annotated with a focus on their involvement in biological processes and signal transduction pathways based on the identified common pathways.

5.5 Detection of Cell Viability

5 × 103 IPEC-J2 cells were cultured in 96-well plated and treated as described above. For the co-treatment of γGC, cells were treated with concentrations of γGC (50-500 μM) and DON exposure. The CCK8 solution (Dojindo Laboratories, Kumamoto, Tokyo, Japan) was dissolved in the medium and then added to each well. Then, cells were co-incubated for 3h. The absorbance at 450 nm was measured using a Tecan Infinite 200 microplate reader (Sunrise model, Tecan, Switzerland).

5.6 Flow Cytometry

Analysis of ROS level was determined using Dihydroethidium (DHE, KeyGEN BioTECH, Jiangsu, China). Each well were incubated with 10 μM of DHE at a temperature of 37℃ for 20 min. After that, cells were washed three times with PBS. The ROS level was detected using a flow cytometer (Excitation wavelengths:488nm, emission wavelengths: 525nm, Beckman Coulter, Brea, USA). To measure the apoptosis rates, cells were treated with Annexin VPE/7-AAD (CA1020, Solarbio, Beijing, China). After staining, a flow cytometer (Excitation wavelengths:488nm, emission wavelengths: 520nm, Beckman Coulter, Brea, USA) was employed to conduct the analysis.

5.7 Western Blotting

Different treated Cells were washed with pre-chilled PBS and lysed with RIPA buffer (Applygen, Beijing, China) with proteinase and phosphatase inhibitors (CWBIO, Beijing, China) on ice for 20 min. Proteins were then separated by SDS-PAGE and transferred to PVDF membranes (Bio–Rad, Hercules, USA). Membranes were blocked with 5% skimmed milk for 2 h and incubated with primary antibodies (BAX, BCL2, 1:5000) at 4°C overnight. After three washes with TBST, membranes were incubated with HRP-conjugated secondary antibodies (1:5000) at room temperature for 2 h. Protein bands were visualized using ECL and analyzed with ImageJ software. Anti-Bax (ER0907) and anti-Bcl2 (SZ10-03) were purchased from Hangzhou HuaAn Biotechnology (Hangzhou, China). HSP90 (60318-1-lg) was used as a loading control and obtained from Proteintech Ltd. (Wuhan, China).

5.8 Statistical Analysis

All data are presented as mean ± standard derivation (SD). Student’s t-test or one-way ANOVA was performed to compare the statistical differences between different groups. The post-hoc tests was used to differentiate significance among different groups. A significance level of p< 0.05 was used to indicate statistical significance:*p <0.05, **p < 0.01.

Round 2

Reviewer 3 Report

Comments and Suggestions for Authors

For the most part, the authors have made the requested changes to the manuscript, particularly concerning the Discussions and the revision and addition of the Bibliography.